# Neoadjuvant Immunotherapy for Muscle-Invasive Bladder Cancer

**DOI:** 10.3390/medicina57080769

**Published:** 2021-07-29

**Authors:** Arthur Peyrottes, Idir Ouzaid, Gianluigi Califano, Jean-Francois Hermieu, Evanguelos Xylinas

**Affiliations:** 1Department of Urology, Bichat-Claude Bernard Hospital, Assistance-Publique-Hôpitaux de Paris, Université de Paris, 75018 Paris, France; arthur.peyrottes@aphp.fr (A.P.); idir.ouzaid@aphp.fr (I.O.); gianl.califano2@gmail.com (G.C.); jean-francois.hermieu@aphp.fr (J.-F.H.); 2Urology Unit, Department of Neurosciences, Reproductive Sciences and Odontostomatology, Federico II University of Naples, 34102 Naples, Italy

**Keywords:** bladder cancer, neoadjuvant, immune checkpoint inhibitors, cystectomy, muscle-invasive bladder cancer

## Abstract

*Background and Objectives*: Facing neoadjuvant chemotherapy followed by surgery, neoadjuvant immunotherapy is an innovative concept in localized muscle-invasive bladder cancer. Herein, we performed a review of the available and ongoing evidence supporting immune checkpoint inhibitor (ICI) administration in the early stages of bladder cancer treatment. *Materials and Methods*: A literature search was performed on Medline and clinical trials databases, using the terms: “bladder cancer” OR “urothelial carcinoma”, AND “neoadjuvant immunotherapy” OR “preoperative immunotherapy”. We restricted our investigations to prospective clinical trials evaluating anti-PD-(L)1 and anti-CTLA-4 monoclonal antibodies. Data on efficacy, toxicity and potential biomarkers of response were retrieved. *Results*: The search identified 6 ICIs that were tested in the neoadjuvant setting for localized bladder cancer—4 anti-PD-(L)1 inhibitors (Pembrolizumab, Atezolizumab, Nivolumab and Durvalumab) and 2 anti-CTLA-4 inhibitors (Ipilimumab and Tremelimumab). Most of the existing literature was based on single-arm phase 2 clinical trials that included from 23 to 143 patients. The pathological complete response rate (pCR) and pathological response rate (pRR) ranged from 31% to 46% and from 55.9% to 66%, respectively. Survival data were immature at this time. The safety profile was acceptable, with severe treatment-related adverse events ranging from 6% to 41%. *Conclusions*: The results of early phase trials are encouraging, and more investigations are needed to strengthen the rationale for immune checkpoint inhibitor administration in localized muscle-invasive bladder cancer.

## 1. Introduction

Bladder cancer accounts for 90% to 95% of urothelial carcinoma, with invasion of the muscle wall accounting for 30% [1]. The current standard of care for localized bladder cancer (T2-T4N0M0) is radical cystectomy, preceded by 6 cycles of neoadjuvant cisplatin-based chemotherapy. Despite the latter therapeutic strategy, outcomes are poor, with an overall added survival benefit of 5–8% at 5 years with chemotherapy administration [1]. Moreover, at the time of diagnosis, approximately half of these patients were ineligible for receiving cisplatin, mostly due to chronic kidney disease (creatinine clearance < 60 mL/min) and performance status (ECOG > 1). For those eligible for cisplatin administration and who do receive the systemic treatment, the toxicity is substantial (renal failure, hearing loss, neuropathy, or heart failure); therefore, this weights the balance between the benefit (moderate) and risk of neoadjuvant chemotherapy. Finally, the inability to predict responders versus non-responders to chemotherapy (thus delaying surgery in the latter) explains a little further the low acceptance of this therapeutic strategy by the uro-oncology community worldwide.

The PD-1 pathway has an important role in local immunosuppression in the tumor microenvironment, but it can also have a role in the modulation of T cell priming. In recent years, immune checkpoint inhibitors (ICIs) targeting programmed cell death, regarding protein-1 (PD-1), programmed death-ligand 1 (PD-L1) and cytotoxic T-lymphocyte-associated protein 4 (CTLA4), have become another pillar of cancer treatment. Concerning urothelial carcinoma, they have shown durable responses in locally advanced and metastatic patients, at first in a second-line setting [2], and more recently in first-line applications, either as a primary treatment or as maintenance after first-line chemotherapy administration. Some of them are also approved as a first-line treatment in cisplatin-ineligible patients [3,4]. As a result, ICTs are being more and more tested in earlier phases of the disease. Herein, we summarize the available published or communicated evidence on neoadjuvant ICIs in non-metastatic muscle-invasive bladder cancer.

## 2. Materials and Methods

We conducted a comprehensive review, using a literature search for current data on perioperative ICIs, using the PubMed and ClinicalTrials.gov databases. The following terms were used: “bladder cancer” OR “urothelial carcinoma”, AND “neoadjuvant immunotherapy” OR “preoperative immunotherapy”. Only prospective studies evaluating ICIs in the neoadjuvant setting in localized bladder cancer were included. Studies were excluded if they involved non-muscle invasive bladder cancer (NMIBC), locally advanced, or metastatic MIBC. Other exclusion criteria were a lack of proven diagnosis, efficacy or safety, and the absence of key information, such as hazard ratios, confidence intervals and *p*-values. In addition, as immunotherapy is a rapidly developing field, we examined abstracts from major oncology conferences until ASCO-GU 2021 (February 2021). We performed our literature search covering the period from 1 January 2010 to 1 March 2021. Three independent authors (A.P., I.O. and E.X.) performed the screening of titles, abstracts, and full-text sharing, and accepting the final list of publications that would be included.

## 3. Results

### 3.1. Evidence Synthesis

#### 3.1.1. Studies Included

The search identified 6 ICIs that were tested in the neoadjuvant setting for localized bladder cancer: 4 anti-PD-(L)1 (Pembrolizumab, Atezolizumab, Nivolumab and Durvalumab) and 2 anti-CTLA-4 (Ipilimumab and Tremelimumab). Most of the existing literature was based on single-arm phase 2 clinical trials, as summarized in Table 1. The sample sizes of these studies ranged from 23 to 143 patients. It is noteworthy that the inclusion criteria were quite similar among trials of cT2-T4a urothelial carcinoma of the bladder without distant metastasis, being mainly a performance status maximum of 2 and no contra-indication to the administration of ICIs. Interestingly, most of the patients were ineligible for cisplatin-based neoadjuvant chemotherapy. Only 1 trial included patients with clinical lymph node involvement [5]. The patients’ characteristics were similar: a high prevalence of male patients, with a median age of 68 years old. Most of the trials used as a primary endpoint a pathological complete response rate (ypT0), similar to that which is used for neoadjuvant chemotherapy evaluation. Since downstaging to <ypT2 might be a marker of response and survival in muscle-invasive bladder cancer [6], the pathological response rate (ypT0/Tis/Ta/T1) was a frequent secondary endpoint. The cancer-specific and overall survival were rarely available due to lack of follow-up, except in the PURE-01 study, which is also the study with the most hindsight [7]. Safety profile and biomarker analyses were also common secondary endpoints.

#### 3.1.2. Oncological Outcomes

Oncological outcomes among the studies are displayed in Table 2. Although study populations were similar, substantial variations were observed concerning the pathological complete response rate (pCR) and pathological response rate (pRR). The latter ranged from 31% to 46% and 55.9% to 66%, respectively, whereas cisplatin-based chemotherapy reported residual high-risk disease (≥ypT2) in more than 50% of the patients [8]. Neoadjuvant immunotherapy might be at least as efficient as chemotherapy regarding pathological response. It is to be noted that most of the ICIs were tested in cisplatin-ineligible patients, skewing the results in favor of monoclonal antibodies. The short follow-up prevents reliable survival estimations, but several trials reported promising findings. With a median follow-up of 23 months in the PURE-01 study, patients being 24-month event-free and their overall survival rates were 71.7% (62.7–82%) and 91% (85.4–97.1%), respectively. Atezolizumab showed a 1-year recurrence-free survival rate of 79% (67–87%), while the association of Durvalumab and Tremelimumab showed 1-year recurrence-free and overall survival rates of 82.8% and 88.8%, respectively [9]. Subgroup analyses based on PD-L1 expression were discordant. The pCR rates seem to be higher in PD-L1-expressing patients [5,10,11], while recurrence-free survival was not impacted [11].

#### 3.1.3. Safety Outcomes

ICIs have mainly been tested in the neoadjuvant setting in cisplatin-ineligible patients. Despite their demonstrated efficacy, ICIs come with a certain degree of potential toxicity. A wide range of side effects has been observed in the presented trials, with fatigue and skin rashes being the most common. Severe treatment-related adverse events (grade 3 or 4 from the National Cancer Institute’s common terminology criteria for adverse events) ranged from 6% to 41% among studies, thus leading to discontinuation or delaying immunotherapy administration in some cases. To our knowledge, no death was reported. The highest rates of adverse events were seen when ICIs were combined. Thus, the associations of Nivolumab + Ipilimumab and of Durvalumab + Tremelimumab reported 41% and 21% of severe AEs rates, respectively, while monotherapies claimed to have less than 10% of severe toxicity. Moreover, the association of several ICIs seems to emphasize side effects without increasing the pathological response rate. Overall, monotherapy with ICIs is usually well tolerated. In the recent PURE-01 trial, thyroid dysfunction was the most frequent all-grade adverse event (AE) (18%). Grade 3 AEs were observed in 6% of patients, and discontinuation of Pembrolizumab was necessary for one of the fifty enrolled patients [10]. Results from the ABACUS trial showed a favorable safety profile: asthenia (18%) was the most common grade 1–2 AE, and transaminitis (4%) was the grade 3–4 most common toxicity [11]. However, combinations may add more toxicities, and careful consideration of the implications of AE should be included. Globally, early-phase trials have shown that neoadjuvant ICIs prior to surgery were well tolerated, with the majority proceeding to radical cystectomy with no delay. Importantly, up to 14% of patients may experience retarded immune-related AEs after cystectomy, as shown recently in an updated analysis of ABACUS, including adrenal insufficiency, hypothyroidism, and an increase in liver enzymes [12]. Thus, monitoring patients for these toxicities is mandatory. Unfortunately, no HRQoL data were yet analyzed in most clinical trials, an endpoint that should be considered in trials aiming to change the standard of care.

#### 3.1.4. Biomarkers for Selection of Patients

With such a craze for PD-(L)1 and CTLA-4 monoclonal antibodies, there is a real need for selecting good responders to immunotherapy. All previous trials have analyzed a panel of biomarkers in order to distinguish those people who will benefit the most from ICIs. PD-L1 expression on surgical specimens was the most studied biomarker and has been positively correlated with response in the PURE-01 and NABUCCO trials, whereas MDACC did not find any statistical association. The tumor mutational burden was also assessed with a cut-off placed between 10 and 15 Mut/Mb and was found to be associated with pT0 response [5,10]. It is noteworthy that PD-L1 expression was not correlated with a high TMB. Another line of research regarded alterations in DNA damage repair (DDR) genes but failed to predict pathological response in several trials. In order to predict the antitumor activity of immune checkpoint inhibitors, a set of immune-gene signatures in pT0 surgical specimens were compared to those ≥pT2. Of these, 18 genes were found to be associated with pathological complete response. The informative genes were mainly involved in antigen presentation pathways and adaptative immune response elaboration [10]. Preexisting cytotoxic T-cell activation was correlated with response in many ways [11]. First of all, the high intraepithelial density of CD8+ cells was associated with higher pCR rates, compared to a lack of CD8+ cells. Second, cytotoxic T-cell gene signatures were significantly increased in responders, compared to patients with stable disease and patients who experienced disease recurrence. Finally, high granzyme B (GZMB) staining—playing a role in T-cell activation—was a surrogate marker of activated CD8+ immunity and, hence, may well impact the response to ICIs. Given that B-cell activity and tertiary lymphoid structures improve immunotherapy in melanoma, these two biomarkers have been tested in bladder cancer [13,14]. In the ABACUS trial, the baseline B-cell presence was correlated with non-response, irrespective of cytotoxic T-cell immunity. TLS at baseline did not predict immunotherapeutic response, although TLS dynamics (induction and maturation of composition) were observed in responding patients. At the opposite end of the spectrum, responders to the association of Durvalumab + Tremelimumab presented higher rates of TLS and B-cells at the baseline and high expression of germinal center initiation genes, such as POU2AF1 [9].

## 4. Discussion

Neoadjuvant cisplatin-based chemotherapy combinations for cT2-T4a N0 bladder cancer are the current standard of care strategy for fit patients, having demonstrated an overall survival benefit in phase III trials and in meta-analyses [15]. The SWOG 8710 study randomized more than 317 MIBC patients to three cycles of neoadjuvant methotrexate, vinblastine, doxorubicin, and cisplatin (MVAC), followed by radical cystectomy, compared to radical cystectomy alone [16]. MVAC increased median survival (77 months vs. 46 months, *p* = 0.06) and improved the rate of pCR, defined as pT0N0 (38% vs. 15%, *p* < 0.001). This study defined pCR as a surrogate for the efficacy for neoadjuvant chemotherapy since 85% of pT0 patients were free of disease at five years. Other trials conducted in this area were unable to demonstrate the benefit of neoadjuvant chemotherapy for different reasons (inadequate statistical power, suboptimal chemotherapy, etc.). In the last decade, alternative regimens (GC) and schedules such as dose-dense MVAC (ddMVAC) have also been tested in two single-arm studies, showing similar pCR compared to conventional MVAC but within a much shorter period of time. More recently, in a randomized phase-III study, ddMVAC has demonstrated a higher local control rate (pCR 42% vs. 36%; <ypT2pN0: 63% vs. 49%) compared to neoadjuvant CG (*p* = 0.021 and 0.007, respectively), with more severe asthenia and gastrointestinal side effects [17]. However, these data are yet to be confirmed in terms of overall survival. Despite the demonstrated oncological benefit, neoadjuvant chemotherapy remains underutilized. Moreover, recurrence rates remain high, and there is no clear impact on locoregional control and locoregional disease-free survival [18]. Importantly, if cisplatin-based neoadjuvant chemotherapy is not possible because of comorbidities or impaired performance status (PS), neoadjuvant chemotherapy should not be given, and upfront cystectomy is recommended since no evidence exists for carboplatin or other therapies in this setting [19]. Consequently, it is necessary to improve outcomes in MIBC with novel therapies. In the current context, in which ICIs have become a new standard of care for advanced urothelial cancer (despite the challenges seen, such as a high rate of progression and lack of clear biomarkers), a plethora of clinical trials incorporating these agents in the early disease setting, where they could increase cure rates, are being developed.

The emerging data from the different trials exploring the role of ICIs in the neoadjuvant setting of UC are promising. The pCR rates observed might lead to a change in the neoadjuvant scenario and set new treatment standards. However, some questions remain unanswered. Firstly, the identification of patients who are most likely to benefit from ICI administration needs to be investigated. There are no molecular biomarkers that are properly validated for patients’ selection. The anti-tumor effect of immunotherapy does not correlate with the PD-L1 expression status, as PD-L1 low-expression tumors may also respond to ICIs. The neoadjuvant setting is an ideal translational research setting, due to the possibility of pre- and post-treatment evaluation. Secondly, we need mid- and long-term survival data to demonstrate the oncological benefits associated with ICI administration; unlike with neoadjuvant chemotherapy, we do not yet have evidence that pCR is a surrogate of overall survival. Moreover, the currently available data come from single-arm phase 2 trials; we need randomized controlled trials to assess the benefits associated with ICIs administration compared to neoadjuvant chemotherapy in all-comers, and also versus upfront surgery in cisplatin-ineligible patients. Thirdly, we need to determine the optimal setting between a neoadjuvant regimen or the combination of a neoadjuvant and maintenance post-surgery regimen, as in the first-line metastatic setting [4].

## 5. Conclusions

Neoadjuvant cisplatin-based chemotherapy has been the standard of care for muscle-invasive bladder cancer for decades. With outstanding results in locally advanced and metastatic urothelial carcinoma, the place of ICIs is being moved forward to earlier stages of the disease. Several phase-2 trials have tested ICIs in localized bladder cancer in the neoadjuvant setting, with promising results and acceptable toxicity. Predictive biomarkers of response are yielded to improve the selection of patients for these new treatment strategies. In order to incorporate them in daily practice, ICIs need randomized controlled trial evidence and longer follow-ups to be implemented as a new standard of care in localized muscle-invasive bladder cancer.

## Figures and Tables

**Table 1 medicina-57-00769-t001:** Available published or communicated studies on neoadjuvant ICIs in muscle-invasive non-metastatic bladder cancer.

Trial Name	Phase	Drug	Target	Regimen	Eligible Patients	Endpoints	Sample Size	PD-L1 Positive
PURE-01	2 (single arm)	Pembrolizumab	PD-1	3 cycles	Eligible and ineligible for cisplatin regimen	pcRR (ITT), EVS, RFS, OS, biomarker analysis, AE	143	35 (70%)
ABACUS	2 (single arm)	Atezolizumab	PD-L1	2 cycles	Ineligible for cisplatin regimen	pcRR, ReFS, Safety, biomarker analysis	88	35 (40%)
NABUCCO	2 (single arm)	Nivolumab-Ipilimumab	PD-1/CTLA-4	2 cycles	Ineligible or refused cisplatin-based chemo	Feasibility to resect within 12 weeks, AE, pcRR	24	15 (63%)
HOG GU 114-88	2	Pembrolizumab-Gemcitabine+/-Cisplatin	PD-1	5 cycles	Eligible for cisplatin regimen	≤pT1, safety, 12-month relapse-free survival, 12-month disease-specific survival, 12 month and 24-month overall survival	80	56%
BLASST	2	Nivolumab-Gemcitabine-Cisplatin	PD-1	4 cycles	Eligible for cisplatin regimen	≤pT1, safety of GC + N, PFS at 2 years, biomarker analysis	41	NA
DUTRENEO	2	Durvalumab-Tremelimumab	PD-L1/CTLA-4	3 cycles	Ineligible for cisplatin regimen	pcRR, OS, disease-free survival, AE, biomarker analysis	23	14 (61%)
MDACC	2 (single arm)	Durvalumab-Tremelimumab	PD-L1/CTLA-4	2 cycles	Ineligible for cisplatin regimen	Safety, pcRR, pRR (downstaging to pT1 or less)	28	NA

pcRR = pathological complete response rate, pRR = pathological response rate, EFS = event-free survival, RFS = recurrence-free survival, ReFS = relapse-free survival, OS = overall survival, AE = adverse events.

**Table 2 medicina-57-00769-t002:** Oncological outcomes of published or communicated studies on neoadjuvant ICIs in muscle-invasive non-metastatic bladder cancer.

Trial Name + B20:J26	Treatment	cT2 Stage	cT3-T4 Stage	cN+ Stage	pT0N0 Rate (cpRR)	pT0N0 Rate (cpRR) in PD-L1 Positive Population	*p* ≤ T1N0 Rate	1-Year RFS	1-Year RFS in PD-L1 Positive Population	1-Year OS
PURE-01	Pembrolizumab	70 (49%)	73 (51%)	0	55 (38.5%, 95% IC: 30.5–46.5%)	54.30%	80 (55.9%, 95% IC: 47.4–64.2%)	84.5% (95% IC: 78.5–90.5%) (Event-free survival)	NA	91% (95% IC: 85.4–97.1%) (2 years OS)
ABACUS	Atezolizumab	64 (73%)	24 (27%)	0	27 (31%, 95% IC: 21–40%)	37%, 95% IC: 21–55%	NA	79% (95% IC: 67–87%) (1-year relapse-free survival)	75–95% IC: 53–87%	NA
NABUCCO	Nivolumab-Ipilimumab	0	14 (58%)	10 (42%)	11 (46%, 95% IC: 26–67%)	73%, 95% IC: 45–92% (pRR)	58%, 95% IC: 37–77%	NA	NA	NA
HOG GU 114-88	Pembrolizumab-Gemcitabine+/-Cisplatin	47%/43.2%	/56.7%	0%/0%	44.4%/45.2%	NA	61.1%/51.6%	/67%	NA	/88.4%
BLASST	Nivolumab-Gemcitabine-Cisplatin	90%	10%	3%	34%	NA	27 (66%)	NA	NA	NA
DUTRENEO	Durvalumab-Tremelimumab	18 (78%)	5 (22%)	8.70%	8 (35%)	57%	13 (56.5%)	NA	NA	NA
MDACC	Durvalumab-Tremelimumab	12 (43%)	15 (54%)	0	9 (37.5%) (pT0 or pTis)	NA	14 (58.3%)	82.8% (1-year relapse-free survival)	NA	88.80%

## Data Availability

Not applicable.

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
