# Peer review of "Neoadjuvant Immunotherapy for Muscle-Invasive Bladder Cancer"

_medicina, 2021, doi:10.3390/medicina57080769_

Round 1
Reviewer 1 Report
The manuscript “Neoadjuvant immunotherapy for localized bladder cancer” by Arthur Peyrottes et al. is well conducted and overall adds important information regarding neoadjuvant immunotherapy as a novel concept in localized muscle-invasive bladder cancer.
Only minor revisions have to be performed.
- In the Introduction section mechanisms of action and relevance of immune check point inhibitors for cancer treatment is missing.
- Data in Table 1 needs substantial reorganization and simplification, especially regarding applied “drug doses” and explanation of “eligible MIBC patients”. In contrast, my suggestion is further stratification of end-points into several groups
Author Response
We thank the reviewer for the time he spent on our manuscript and his important comments.
- As suggested by the reviewer, we added a paragraph in the Introduction section highlighting the mechanisms of action and relevance of immune check point inhibitors for cancer treatment.
- Moreover, we improved the Table 1 according to the comments.
Reviewer 2 Report
This systematic review summarizes the outcomes of neoadjuvant immune checkpoint inhibitors for localized muscle-invasive bladder cancer
1) The title is "Neoadjuvant immunotherapy for localized bladder cancer", but this title appears to include non-muscle-invasive bladder cancer and to be misleading.
2) Does this systematic review follow the PRISMA checklist? I think it is necessary to present the flow diagram which shows the process of systematic review including searches of databases.
3) The safety outcomes of neoadjuvant immunotherapy should be summarized in a table.
4) There are some errata in this paper. For example, in line 17, it is described that "We r stricted ....". Moreover, both of the 3.1.3. and 3.1.4. subtitles are "Safety outcomes". I think the 3.1.4. subtitle is about biomarkers to select patients for neoadjuvant immunotherapy. The authors should proofread the paper before resubmission.
Author Response
We thank the reviewer for the time he spent on our manuscript and his comments that improved the manuscript.
1) We changed the title in order to be more clear.
2) We conducted a comprehensive review of the available data of published and orally presented clinical trials on neoadjuvant immunotherapy.
3) We added a paragraph on safety/toxicity information of these drugs as suggested by the reviewer.
4) We corrected the mistakes underlined by the reviewer.
Round 2
Reviewer 2 Report
The process of systematic review should be described. Is this study in accordance with the PRISMA guideline? The PRISMA flow diagram should be added.